# Unveiling Causal Relationships Among Candidate Output Tokens in Large Language Models: Towards Interpretability and Control

## Abstract

Understanding how large language models (LLMs) generate tokens is crucial for enhancing their performance and interpretability. We hypothesize that cause-effect relationships exist among candidate output tokens during next token prediction in LLMs. Specifically, we propose that certain candidate output tokens—termed "effect tokens"—are causally influenced by other candidate tokens activated in earlier layers, referred to as "cause tokens". To test this hypothesis, we develop a causal analysis methodology that uncovers these relationships within open-source LLMs. We find that while cause tokens are essential for generating effect tokens, including them in the final output can degrade model performance.

Building on these findings, we introduce a decoding algorithm that employs two heuristics: Critical Layer Ablation (CLA), which approximates causal relationships by selectively removing transformer layers and observing their impact on token generation, and Causally-Informed Decoding (CID), which uses the relationships identified by CLA to adjust token probabilities. Specifically, CID increases the probability of selecting effect tokens while decreasing that of cause tokens during generation. Our method achieves measurable accuracy improvements across various benchmark datasets, demonstrating its potential to enhance both the controllability and performance of LLM-generated text.

## 1 Introduction

Large language models (LLMs) have achieved impressive performance in natural language processing tasks, attracting significant attention from academia and industry (OpenAI, 2022; Dubey et al., 2024). Researchers are increasingly interested in understanding the activation dynamics within LLMs and how it influences the generation process (Edunov et al., 2019; Li et al., 2024). However, the internal mechanisms of LLMs remain largely opaque, posing challenges for interpretability and trustworthiness.

Previous work has investigated the dynamics of LLMs across different layers. Tenney et al. (2019) found that different layers in an LLM serve distinct functions–early layers handle basic linguistic information such as grammar, while later layers capture broader contextual relationships and are thus better at reasoning. Therefore, it is possible that tokens activated[1] in earlier layers differ from those activated in later layers. Furthermore, modern LLMs consist of multiple transformer layers (Vaswani, 2017) with residual connections, which were introduced to stabilize training in deep models (Liu et al., 2020). Each layer transforms its input into a residual "correction" that is then added to the input. This means that tokens activated in early layers—even if they are incorrect or suboptimal—can persist and influence the final output, possibly appearing with high probability.

This creates a *fascinating paradox*: tokens that appear incorrect in earlier layers may not only be crucial stepping stones for generating accurate tokens in later layers but also persist in the final output due to residual connections. Fig. 1 illustrates this phenomenon, where the initial token "Yes",

---

[1]In the context of LLMs, activating a token refers to the process where a layer modifies the hidden embedding, causing the token's projected logit (i.e., the logit obtained by applying the output head directly to the internal hidden embedding) to become prominent.

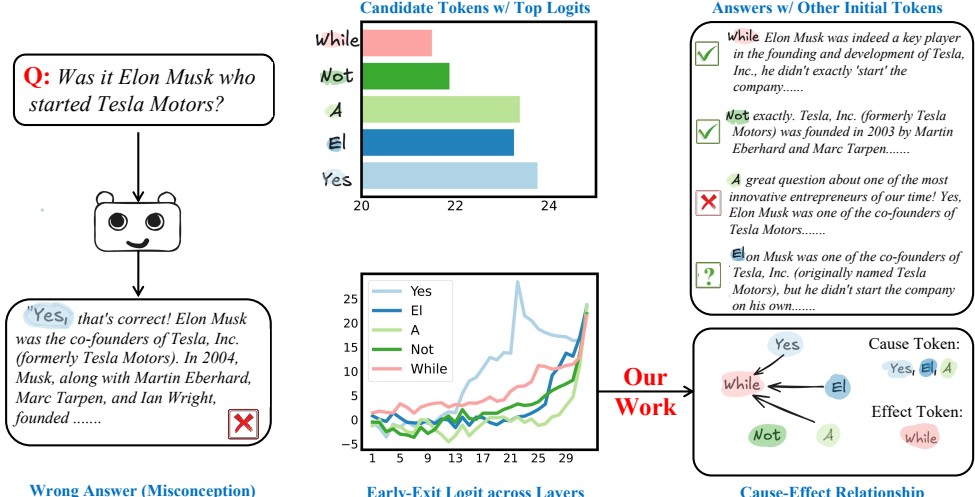

Figure 1: A case example of the cause-effect (CE) relationship between candidate output tokens in Llama-3.1-8B-Instruct, using a TruthfulQA question (Lin et al., 2021). The candidate token "Yes", while factually incorrect, exhibits a high logit value early in the LLM's processing. In contrast, the correct tokens, "While" and "Not", which lead to more nuanced and accurate answers, activate in later layers. Our analysis reveals that the activation of "While" is causally influenced by the initial activation of "Yes".

which leads to an incorrect answer, is activated in early layers and subsequently influences the more nuanced token "While" in deeper layers. The final logit values of both tokens remain high.

With these insights, we hypothesize that during token generation, certain candidate output tokens *causally result* from the activation of others. We term the anticedent tokens "cause tokens" and the subsequent ones "effect tokens." In the example of Fig. 1, "Yes" is a cause token and "While" is an effect token. The logit values for both tokens are tracked across layers, highlighting how early activations can influence later outputs. This example underscores the complex causal dynamics within LLMs, where cause tokens can trigger the activation of more contextually appropriate effect tokens in deeper layers.

We propose a novel approach to test the hypothesis that cause-effect relationships exist among candidate output tokens during inference. Recognizing that, due to residual connections, the transformations from all layers are ultimately *stacked into the final layer's output*, we simplify causal analysis by focusing directly on this final output and constructing the causal graph based on the candidate output tokens themselves. This direct construction of causal graph, as elaborated in Section 3, enables a clear and intuitive understanding of the relationships among these tokens. In contrast, previous works have often tackled the more intricate task of analyzing the causal graph across the entire neural network.

As a second major contribution, inspired by our understanding of the causal relationships among candidate output tokens, we propose to leverage this knowledge to enhance the decoding process. Our aim is to guide the LLM towards selecting tokens that better align with the discovered causal dependencies. Specifically, we propose to increase the probabilities of choosing effect tokens, while decreasing the probabilities of choosing their corresponding cause tokens. However, directly integrating full-scale causal analysis into decoding is computationally prohibitive. To address this challenge, we introduce the Critical Layer Ablation (CLA) heuristic, a method designed to efficiently approximate causal relationships in real-time text generation scenarios.

Building upon the causal discoveries by CLA, we introduce a novel decoding algorithm, Causally-Informed Decoding (CID). CID dynamically modifies logit values prior to sampling the output token, incorporating causal considerations to enhance the generation process. CID significantly improves the models' reasoning performance under the zero-shot and zero-shot chain-of-thought settings across multiple arithmetic benchmarks.

## 2  RELATED WORK

### 2.1  CAUSAL ANALYSIS FOR LLM INTERPRETABILITY

The investigation of reasoning mechanism within (large) language models has garnered increasing attention. The most pertinent series of work to ours adopted a causal analysis tool, called *causal mediation analysis* (PEARL, 2001) or *activation patching* (Zhang & Nanda, 2024) in some context. Causal mediation analysis detects which part of a neural network is responsible for the correct reasoning over a specific input sample through *intervention*, which corrupts the prediction by perturbing the input and restores it by *patching* the activations from the clean run. This methodology was first introduced to language modeling by Vig et al. (2020) for gender bias investigation. Following work has then focused on its application to different tasks, including subject-verb agreement (Finlayson et al., 2021), natural language inference (Geiger et al., 2021), indirect object identification (Wang et al., 2022a), multiple-choice question answering (Wiegreffe et al., 2024) and syllogistic reasoning (Kim et al., 2024).

These prior works differ from ours in their focus on internal model representations. We, however, shift the focus to the causal dynamics among the output tokens themselves. We acknowledge that while analyzing activations within internal layers is intuitive, residual connections in LLMs allow earlier activations' causal influence to persist, impacting the final output. Therefore, we aim to uncover cause-effect relationships directly among candidate tokens, offering a unique perspective.

### 2.2  LOGIT- AND DECODING-LEVEL REASONING

In addition to causal analysis, research has explored reasoning at the token and decoding levels in LLMs. Wang et al. (2022b) proposed self-consistency checking during Chain-of-Thought decoding to enhance coherence. Chuang et al. (2023) introduced DoLa, modifying logits by differentiating between layers to improve text quality and diversity. These techniques, while valuable, do not explicitly consider underlying causal relationships among tokens. Our work complements these efforts by introducing a causally-informed decoding algorithm that leverages identified cause-effect relationships to further enhance accuracy and faithfulness to the model's internal reasoning. By integrating causal analysis with token-level reasoning, we aim to bridge the gap between theoretical understanding and practical control of LLMs, paving the way for more controllable, reliable, and causally-grounded text generation.

### 2.3  ADDITIONAL RELEVANT WORKS

Rajani et al. (2019) presented an early yet impactful work on causal reasoning in language models, emphasizing the importance of understanding cause-effect relations. Feder et al. (2021) explored the integration of causal graphs into language models, fine-tuning deep contextualized embedding models with auxiliary adversarial tasks. Jin et al. (2023) introduced a novel task of causal inference in natural language, accompanied by a dataset comprising causal graphs and queries. Chen et al. (2024) proposed a new causal evaluation framework, offering a systematic approach to assess causal reasoning abilities.

## 3  CAUSAL ANALYSIS FORMULATION

We consider an LLM built with $n$ transformer layers, focusing specifically on how it generates a single output token given a fixed text input. Let $Y = \{1, 2, ..., m\}$ represent the set of all possible output tokens (the vocabulary), where $m$ is the vocabulary size. Our goal is to uncover causal relationships among elements in $Y$. To achieve this, we focus on analyzing the final output logits on $Y$, constructing a causal graph based on the candidate output tokens themselves. We elaborate on the details of this causal graph construction in Section 3.1.

Let $\mathbf{e} \in \mathbb{R}^d$ be the initial embedding of the input text, where $d$ is the hidden dimension.

Each transformer layer $l = 1, 2, ..., n$ acts as a transformation function $\Delta^{(l)} : \mathbb{R}^d \to \mathbb{R}^d$. This function takes the input representation at layer $l$ (the output from layer $l - 1$) and computes the

change, or delta, in the representation at layer $l$. This delta reflects the combined impact of the layer's attention mechanisms and feedforward networks.

Let $\mathbf{h}^{(l)} \in \mathbb{R}^d$ represent the embedding of the output in the layer $l$, for $l = 1, 2, \ldots, n$. In a typical LLM setup, the final embedding of the output after the $n$-th layer is calculated as:

$$\mathbf{h}^{(n)} = \mathbf{e} + \sum_{l=1}^{n} \Delta^{(l)}(\mathbf{h}^{(l-1)}),$$

where $\mathbf{h}^{(0)} = \mathbf{e}$ is the initial embedding.

Let $L : \mathbb{R}^d \setminus \{\mathbf{0}\} \to \mathbb{R}^d$ denote a normalization layer that takes a vector and returns its normalized version scaled by a constant:

$$L(\mathbf{v}) = \frac{\mathbf{v}}{\|\mathbf{v}\|_2} \cdot \gamma,$$

where $\gamma$ is a scalar and $\|\mathbf{v}\|_2$ is the 2-norm (Euclidean norm) of $\mathbf{v}$. Let $\mathbf{H} \in \mathbb{R}^{m \times d}$ be the head matrix. Then the product

$$\mathbf{H} \cdot L(\mathbf{h}^{(n)})$$

gives us the logit values for each candidate token in $Y$.

Past research has used intervention techniques to study how internal transformations $\Delta^{(l)}(\cdot)$ affect the final output $\mathbf{h}^{(n)}$. Our approach is distinct: We directly examine the causal graph on the final logit values $\mathbf{H} \cdot L(\mathbf{h}^{(n)})$, without analyzing the complex individual layer contributions. This is possible because, as shown in the equation above, the internal transformations $\Delta^{(l)}(\cdot)$ across all layers $l$ are cumulatively integrated to form $\mathbf{h}^{(n)}$.

### 3.1 CONSTRUCTING THE MARKOV EQUIVALENCE CLASS

To uncover the causal relationships among candidate tokens $Y$, we construct the Markov equivalence class (Spirtes et al., 2001), represented by a Completed Partially Directed Acyclic Graph (CPDAG) (Andersson et al., 1997). A Markov equivalence class comprises causal graphs encoding the same conditional independencies, indistinguishable based on observational data. The CPDAG captures both directed and undirected edges, representing identifiable causal relationships and ambiguities.

We generate observational data through deliberate perturbations, a common method in causal discovery that reveals partial causal structures without interventional data. We introduce random scalars $\boldsymbol{\alpha} = \{\alpha_1, \alpha_2, ..., \alpha_n\}$, with each $\alpha_l$ drawn independently from a predefined distribution, to perturb the transformations $\Delta^{(l)}$. This controlled randomness allows us to probe the model's behavior during single-token generation.

Arithmetically, the output representation at layer $l$ now becomes:

$$\tilde{\mathbf{h}}^{(l)} = \tilde{\mathbf{h}}^{(l-1)} + \alpha_l \cdot \Delta^{(l)}(\tilde{\mathbf{h}}^{(l-1)})$$

with $\tilde{\mathbf{h}}^{(0)} = \mathbf{e}$, and the final output at the $n$-th layer is

$$\tilde{\mathbf{h}}^{(n)} = \mathbf{e} + \sum_{l=1}^{n} \alpha_l \Delta^{(l)}(\tilde{\mathbf{h}}^{(l-1)}).$$

For a given distribution of the random scalars, we generate $k$ samples of $\tilde{\mathbf{h}}^{(n)}$, denoted by $\tilde{\mathbf{h}}_1^{(n)}, \tilde{\mathbf{h}}_2^{(n)}, \ldots, \tilde{\mathbf{h}}_k^{(n)}$. Multiplying these by the head matrix after applying the normalization layer gives us a sample of logit value vectors $\hat{S}_{\boldsymbol{\alpha}} = \{\mathbf{s}_1, \mathbf{s}_2, ..., \mathbf{s}_k\}$, where $\mathbf{s}_i \in \mathbb{R}^m$ and

$$\mathbf{s}_i = \mathbf{H} \cdot L(\tilde{\mathbf{h}}_i^{(n)}) = \mathbf{H} \cdot \frac{\tilde{\mathbf{h}}_i^{(n)}}{\|\tilde{\mathbf{h}}_i^{(n)}\|_2} \cdot \gamma.$$

Table 1: Statistics of cause-effect relationships detected by the Peter-Clark algorithm at different significance levels (P-values). 'ce (%)' represents the percentage of candidate tokens participating in cause-effect relationships. 'c/e' denotes the average number of cause tokens per effect token. 'DI (%)' indicates the proportion of directed edges in the Markov equivalence class, reflecting the clarity of causal directionality.

| P-value | 1e-4 | | | 1e-5 | | | 1e-6 | | |
|---------|--------|------|--------|--------|------|--------|--------|------|--------|
| | ce (%) | c/e | DI (%) | ce (%) | c/e | DI (%) | ce (%) | c/e | DI (%) |
| Gemma-2-2B-Instruct | 66.66 | 1.18 | 39.58 | 66.66 | 1.13 | 35.41 | 55.55 | 1.08 | 32.50 |
| Llama-3.2-3B-Instruct | 48.64 | 1.43 | 66.66 | 51.35 | 1.34 | 61.18 | 43.24 | 1.24 | 55.46 |
| Yi-1.5-9B-Chat | 53.00 | 1.22 | 48.84 | 48.63 | 1.20 | 45.36 | 43.71 | 1.15 | 42.18 |
| Llama-3.1-8B-Instruct | 49.02 | 1.28 | 52.56 | 49.67 | 1.26 | 50.40 | 43.18 | 1.21 | 46.42 |
| Gemma-2-9B-Instruct | 70.58 | 1.29 | 45.83 | 64.70 | 1.25 | 44.31 | 47.05 | 1.15 | 35.97 |
| Mistral-Nemo-Instruct | 52.30 | 1.20 | 46.85 | 40.64 | 1.19 | 45.52 | 34.53 | 1.17 | 43.57 |

We employ the Peter-Clark (PC) algorithm[2] (Spirtes et al., 2001) on the sample $\hat{S}_{\alpha}$ to infer the Markov equivalence class represented by a CPDAG $G_{\alpha}$. A directed edge in $G_{\alpha}$ from token $i$ to $j$ signifies that $i$ is a likely cause (cause token) of $j$ (effect token), whereas an undirected edge suggests an uncertain causal direction—the tokens may influence each other.

Our experiments with various LLMs on the GSM8K dataset provide insightful findings into the causal dynamics of token generation, as detailed in Table 1. The data reveals several key observations:

- **High Participation in Causal Relationships ('ce (%)'):** A significant percentage of candidate tokens participate in cause-effect relationships across different models and P-value thresholds, indicating substantial interdependence among tokens during generation.

- **Complex Interactions ('c/e'):** The average number of cause tokens per effect token suggests that effect tokens are influenced by multiple cause tokens, reflecting the complexity of token interactions in LLMs.

- **Prevalence of Directed Edges ('DI (%)'):** A considerable proportion of edges are directed, highlighting prevalent cause-effect relationships with clear directionality among tokens.

These observations underscore the intricate web of causal interactions among tokens in LLMs.

## 3.2 BIAS ANALYSIS

While the introduction of random scalars $\alpha_l$ is crucial for our analysis, it inevitably introduces a degree of bias. To examine this, we construct Markov equivalence classes for various distributions of $\alpha_l$, progressively increasing the probability that $\alpha_l$ will be 1 (minimal perturbation).

We apply the PC algorithm (see Appendix A for a detailed procedure) on three sets of samples obtained by perturbing the LLM on the GSM8K dataset using Bernoulli distributions Bern(0.95), Bern(0.90), and Bern(0.85) for the random scalars $\alpha_l$. We consider the cause and effect tokens derived from Bern(0.95)—which introduces minimal perturbation and hence minimal bias—as the approximate ground truth, and treat the results from Bern(0.90) and Bern(0.85) as predictions. The ROC scatter plot, showing the True Positive Rate (TPR) versus the False Positive Rate (FPR)[3], is presented in Fig. 2. Our findings reveal that as the perturbation distribution concentrates around 1, the constructed Markov equivalence classes (i.e., the discovered causal-effect token pairs) exhibit statistical similarity. This robustness to the choice of perturbation further strengthens the validity of our causal analysis.

---

[2]We utilize the Python package `causal-learn` (Zheng et al., 2024) for implementing and evaluating PC algorithm.

[3]TPR measures the proportion of actual cause-effect pairs correctly identified; FPR measures the proportion of non-causal pairs incorrectly identified.

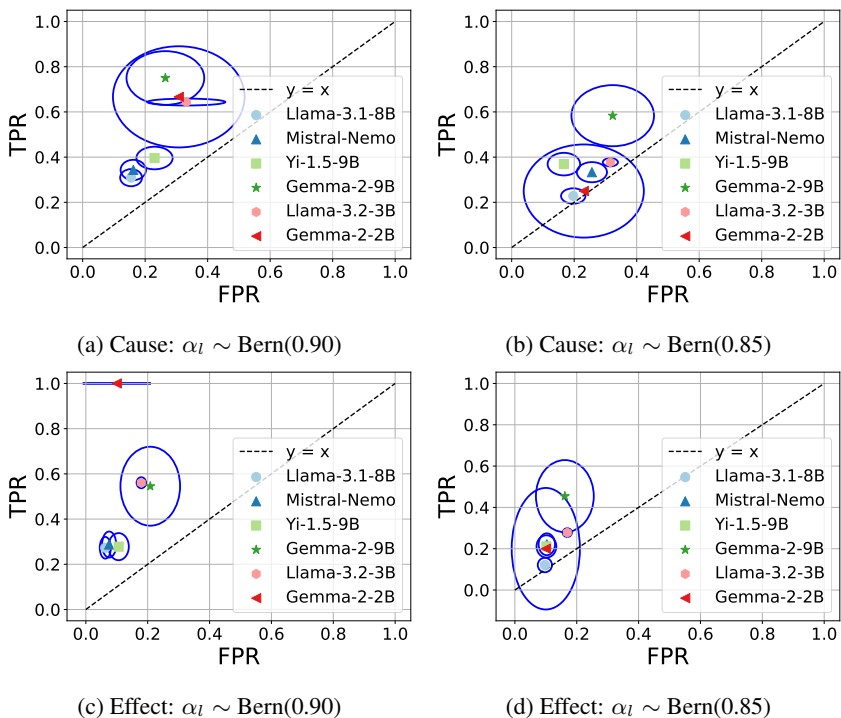

(a) Cause: $\alpha_l \sim \mathrm{Bern}(0.90)$

(b) Cause: $\alpha_l \sim \mathrm{Bern}(0.85)$

(c) Effect: $\alpha_l \sim \mathrm{Bern}(0.90)$

(d) Effect: $\alpha_l \sim \mathrm{Bern}(0.85)$

Figure 2: The ROC scatter plot for bias analysis on GSM8K, with ellipses representing confidence regions. The CE tokens sampled from Bern(0.90) are closer to the ground truth than those from Bern(0.85).

## 4 CAUSALLY-INFORMED DECODING FOR ENHANCED LANGUAGE GENERATION

We introduce Causally-Informed Decoding (CID), a new algorithm that enhances language generation by leveraging causal relationships among the candidate output tokens. To efficiently approximate these causal relationships during decoding, CID employs the novel Critical Layer Ablation (CLA) heuristic, which we detail in the following subsection.

### 4.1 THE CLA HEURISTIC

The CLA heuristic efficiently approximates cause-effect relationships among candidate tokens, significantly faster than the causal discovery algorithms used in Section 3.1. Inspired by intervention techniques, CLA simulates variable manipulation by systematically "ablating" transformer layers and observing the impact on token generation. The core idea is to identify the layer whose transformation has the most significant impact on the logit value of a given token. This layer is considered "critical" for the generation of that token, and its removal can potentially reveal causal dependencies with other tokens.

Specifically, for each token $i \in Y$ among the top candidate tokens at a decoding step, we identify its critical layer $l_i^*$ as the layer that maximizes the relative increment in its logit value:

$$l_i^* = \arg\max_{1 \le l \le n-4} \frac{[\mathbf{H} \cdot L(\mathbf{h}^{(l)})]_i - [\mathbf{H} \cdot L(\mathbf{h}^{(l-1)})]_i}{[\mathbf{H} \cdot L(\mathbf{h}^{(l-1)})]_i} \tag{1}$$

This equation quantifies the relative change in the logit value of token $i$ caused by the transformation at layer $l$. The layer with the maximum relative change is deemed critical for generating $i$. We exclude the last four layers from ablation as they are crucial for producing coherent and contextually relevant generations.

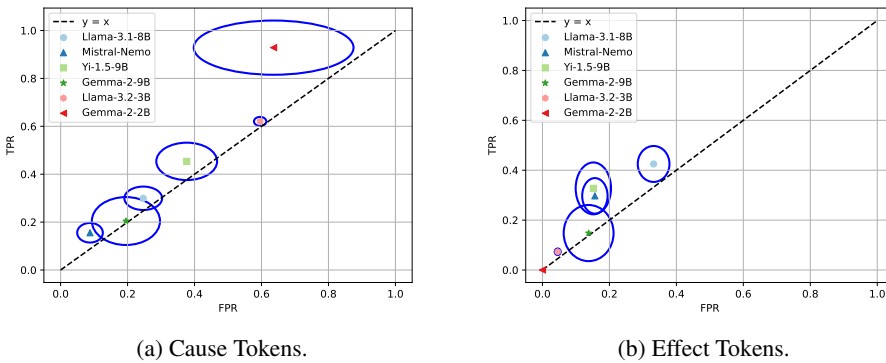

(a) Cause Tokens.                         (b) Effect Tokens.

Figure 3: ROC plot of CLA on the GSM8K.

To simulate the removal of layer $l_i^*$, we modify the computation of the final output embedding $\mathbf{h}^{(n)}$ by simply skipping the transformation at layer $l_i^*$. Let's denote the resulting final output embedding as $\hat{\mathbf{h}}_i^{(n)}$, defined recursively as follows:

$$\hat{\mathbf{h}}_i^{(l)} = \begin{cases} \mathbf{h}^{(l)}, & \text{if } l < l_i^* \\ \mathbf{h}^{(l_i^*-1)}, & \text{if } l = l_i^* \\ \hat{\mathbf{h}}_i^{(l-1)} + \Delta^{(l)}(\hat{\mathbf{h}}_i^{(l-1)}), & \text{if } l > l_i^* \end{cases} \tag{2}$$

If ablating this critical layer also leads to a significant drop in the logit value of another token $j$, it suggests a potential causal dependency from $i$ to $j$.

---

**Algorithm 1** Critical Layer Ablation (CLA) Heuristic for Approximating Causal Relationships

---

**Input:** • The language model (LLM) and the given text input.
      • $d$: Number of top tokens to consider (e.g., $d = 5$)

**Output:** $C$, the set of potential cause-effect pairs detected for the current decoding step.

1: $C \leftarrow \emptyset$
2: $T \leftarrow$ top-$d$ tokens based on the logit values $\mathbf{H} \cdot L(\mathbf{h}^{(n)})$
3: **for** $i \in T$ **do**
4:     Compute $l_i^*$ according to Equation 1
5:     Compute $\hat{\mathbf{h}}_i^{(n)}$ according to Equation 2
6:     $T' \leftarrow$ top-$d$ tokens based on the logit values $\mathbf{H} \cdot L(\hat{\mathbf{h}}_i^{(n)})$
7:     **for** $j \in T$ **do**
8:         **if** $j \notin T'$ **then**
9:             $C \leftarrow C \cup \{(i,j)\}$
10:         **end if**
11:     **end for**
12: **end for**
13: **return** $C$

---

## 4.2 EMPIRICAL VALIDATION OF CLA

To assess the efficacy of CLA in capturing genuine causal relationships, we conduct an empirical validation, comparing the cause-effect pairs identified by CLA against the causal relationships obtained from the Markov equivalence class presented in Section 3.1. We utilize the ROC scatter plot with True Positive Rate (TPR) and False Positive Rate (FPR) to quantify the performance of CLA. Note that TPR ensures the proportion of actual cause-effect pairs correctly identified by CLA, and FPR measures the proportion of non-causal pairs incorrectly identified as cause-effect pairs by CLA. As shown in Fig. 3, CLA's predictions are statistically significant across LLMs.

## 4.3 THE CID ALGORITHM

The CID algorithm leverages causal relationships identified by CLA to modify logits during decoding. CID reduces the probability of sampling cause tokens and increasing the probability of sampling effect tokens. We expect this manipulation of the causal flow to improve the quality of the generated text. Given an input text to the LLM, the CID heuristic operates as described in Alg. 2.

---

**Algorithm 2** Causally-Informed Decoding (CID) for Next Token Prediction

---

**Input:** • The language model (LLM) and the input text.
  • $d$: the number of top tokens to consider
  • $h$: the change in logit value for cause and effect tokens

**Output:** $t \in Y$, the next token generated by the LLM after applying causal modifications.

1:  $\mathbf{Z} \leftarrow \mathbf{H} \cdot L(\mathbf{h}^{(n)})$
2:  $C \leftarrow$ run CLA with parameter $d$
3:  **for** $(i, j) \in C$ **do**
4:      $\mathbf{Z}[i] \leftarrow \mathbf{Z}[i] - h$
5:      $\mathbf{Z}[j] \leftarrow \mathbf{Z}[j] + h$
6:  **end for**
7:  $\mathbf{P} \leftarrow \text{softmax}(\mathbf{Z})$
8:  $t \leftarrow$ sample from $\mathbf{P}$
9:  **return** $t$

---

## 4.4 EXPERIMENTS

In this section, we empirically evaluate the CID algorithm. We conduct experiments across a diverse set of language models and benchmark datasets to assess its impact on the generation quality and logical coherence of the output text.

**Models, Datasets** We utilize two tiny and four small state-of-the-art and representative LLMs to empirically test the performance of the CID algorithm, namely Gemma-2-2B-Instruct (Team et al., 2024), Llama-3.2-3B-Instruct (Touvron et al., 2023), Gemma-2-9B-Instruct (Team et al., 2024), Llama-3.1-8B-Instruct (Touvron et al., 2023), Yi-1.5-9B-Instruct (Young et al., 2024) and Mistral-Nemo-Instruct Jiang et al. (2023). We conduct our experiments on four arithmetic reasoning datasets, namely GSM8K (Cobbe et al., 2021), MAWPS (Koncel-Kedziorski et al., 2016), MultiArith (Roy & Roth, 2016) and SingleEq (Koncel-Kedziorski et al., 2015).

**Settings** We adopt two types of prompts to the LLMs, 'Raw' and 'CoT', where 'Raw' refers to zero-shot question answering without additional prompts, 'CoT' refers to zero-shot prompt with 'Let's think step by step' (Kojima et al., 2022), which is shown to significantly increase models' reasoning capability. We apply two sets of hyper-parameters for the proposed CID algorithm, CID with mild hyper-parameter ($(d, h) = (2,5)$) and CID+ with aggressive hyper-parameter ($(d, h) = (5,10)$).

**Observations:** Table 2 presents the accuracy results of our Causally-Informed Decoding (CID) algorithm compared to the original decoding method across various language models and arithmetic reasoning datasets. We observe substantial gains in accuracy when applying CID to the Yi-1.5-9B-Chat, Mistral-Nemo-Instruct and Gemma-2-2b-Instruct models. For instance, on the GSM8K dataset, accuracies are dramatically improved from 24.30% to 41.30% for Yi-1.5B-Chat and from 12.80% to 43.30% for Mistral-Nemo-Instruct. This suggests that CID is particularly effective in enhancing the reasoning capabilities of these models. While the improvements are less pronounced for Llama-3.1-8B-Instruct and Llama-3.2-3B-Instruct, we still see consistent gains across all datasets. As to Gemma-2-9B-Instruct, CID can at most maintain the performance of this model itself. Note that the CE token prediction performance with CLA on this model is also the worst among all models (see Fig. 3), which might explain why the effect of CID on this model is not particularly significant. These empirical results demonstrate the effectiveness of our Causally-Informed Decoding (CID) algorithm in enhancing the reasoning capabilities of diverse language models across various benchmark datasets. By leveraging causal relationships extracted through the CLA heuristic, CID guides the decoding process towards generating more logically coherent and accurate responses, particularly in challenging reasoning scenarios.

Table 2: Performance of the CID algorithm with different LLMs on arithmetic reasoning datasets. 'Raw' refers to zero-shot question-answering without additional prompts, 'CoT' means zero-shot questions with the prompt 'Let's think step by step'. 'CID+' stands for CID with a more aggressive set of hyper-parameter configuration that encourages more causal influence duing decoding.

| Model | Prompt | Method | GSM8K | MAWPS | MultiArith | SingleEq |
|---|---|---|---|---|---|---|
| Gemma-2-2b-it | Raw | Orig. | 16.30 | 57.98 | 11.00 | 69.69 |
| | | CID | 17.74 | 59.24 | 10.83 | 71.85 |
| | | CID+ | **36.69** | **61.76** | **50.00** | **77.95** |
| | CoT | Orig. | 34.57 | 63.87 | 44.17 | 74.02 |
| | | CID | 24.72 | 65.97 | 51.67 | 74.80 |
| | | CID+ | **35.94** | **78.99** | **77.00** | **85.63** |
| Llama-3.2-3B-Instruct | Raw | Orig. | **71.11** | 83.19 | 66.50 | **90.35** |
| | | CID | 62.02 | **84.87** | 67.33 | 89.96 |
| | | CID+ | 64.97 | 83.61 | **76.17** | 83.56 |
| | CoT | Orig. | 73.09 | **89.92** | **96.00** | 94.49 |
| | | CID | **73.84** | 86.55 | 93.67 | **94.69** |
| | | CID+ | **73.84** | 87.39 | 95.83 | 94.49 |
| Gemma-2-9b-it | Raw | Orig. | **87.34** | **92.86** | **98.33** | **92.52** |
| | | CID | 68.92 | 92.44 | 95.83 | 91.54 |
| | | CID+ | 79.91 | 87.82 | 92.83 | 87.80 |
| | CoT | Orig. | 86.58 | **91.18** | **98.17** | **92.13** |
| | | CID | **86.81** | **91.18** | **98.17** | **92.13** |
| | | CID+ | 82.79 | 90.76 | 90.76 | 91.73 |
| Llama-3.1-8B-Instruct | Raw | Orig. | 79.38 | **92.44** | 93.00 | 92.52 |
| | | CID | **80.89** | 92.02 | 93.67 | 92.13 |
| | | CID+ | 77.86 | 90.76 | **96.50** | **92.72** |
| | CoT | Orig. | 81.12 | 89.08 | 96.83 | 89.76 |
| | | CID | **82.34** | **91.18** | 96.83 | 90.55 |
| | | CID+ | 79.91 | 90.34 | **97.67** | **91.34** |
| Yi-1.5-9B-Chat | Raw | Orig. | 24.72 | 75.21 | 46.67 | 80.51 |
| | | CID | 38.13 | **78.15** | **53.33** | **81.69** |
| | | CID+ | **42.15** | 71.85 | 48.00 | 78.74 |
| | CoT | Orig. | **83.77** | 93.28 | **97.50** | 94.49 |
| | | CID | 83.24 | **94.12** | 97.17 | **94.69** |
| | | CID+ | 82.34 | 92.86 | 96.83 | 94.49 |
| Mistral-Nemo-Instruct | Raw | Orig. | 13.19 | 67.23 | 28.67 | 79.33 |
| | | CID | 19.71 | 68.49 | 28.00 | 79.72 |
| | | CID+ | **45.26** | **71.43** | **48.00** | **84.06** |
| | CoT | Orig. | **69.29** | 77.31 | 81.50 | 87.01 |
| | | CID | 64.82 | 76.05 | 83.00 | 87.40 |
| | | CID+ | 62.09 | **83.61** | **83.67** | **87.99** |

## 5  CONCLUSION AND LIMITATIONS

In this paper, we first hypothesize and verify the CE relationship between candidate output tokens. Inspired by the CE relationship, we designed the CLA method to more efficiently explore the CE connections between tokens and proposed the CID decoding algorithm to enhance the reasoning capabilities of LLMs. Across multiple arithmetic datasets, CID significantly improved the performance of various LLMs.

In this work, the experiments are focused on comparatively smaller LLMs and arithmetic reasoning datasets. Future work will explore the applicability of CID to larger LLMs (more than 70 billion parameters) and other language generation tasks, and will investigate the factors influencing its performance across different model architectures and dataset characteristics.

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

## A    DETAILS OF PETER-CLARK ALGORITHM

Here, we provide more details of the Peter-Clark (PC) algorithm (Spirtes et al., 2001) and how we apply it to our scenarios. Given the input as a tensor $T \in \mathbb{R}^{n \times k}$, which contains the logit value of $n$ candidate tokens repeatedly sampled $k$ times. PC algorithms output a causal graph among each candidate variable via independent tests as follows:

1. We repeatedly perturb the LLM to generate $k = 1000$ samples of logit values for the candidate tokens, $\{s_1, s_2, \ldots, s_k\}$. Each time,the LLM is perturbed by applying Bernoulli random scalars with success probability 0.95 for the layers as described in Section 3.1, effectively removing some of the transformer layers.

2. We then apply the PC algorithm to the generated samples $\{s_1, s_2, \ldots, s_k\}$ using Fisher's z-test for independence with a significance level of $10^{-4}$. This is done using the causal-learn package (Zheng et al., 2024) as footnoted on page 5. The PC algorithm outputs a causal graph between the tokens.

3. Finally we convert the source and destination tokens of each directed edge of the causal graph to a cause-effect pair.

## B    EXPERIMENTS ON ADDITIONAL TASK AND COMPARISON WITH DOLA

As suggested by Reviewer LYVs, we conduct two sets of additional experiments to validate the effectiveness of CID.

**Comparison against DoLa**  DoLa (Chuang et al., 2023) is a new decoding strategy which modifies token logits based on the contrasting between intermediate logits. As CID also improves decoding by modifying token logits, it is natural to compare CID against DoLa. We apply DoLa to the Mistral-Nemo-Instruct model across all four datasets used in Section 4.4. We adopt the recommended settings for long-answer reasoning tasks, such as GSM8K, as suggested by the authors of DoLa: applying DoLa to lower layers and setting the repetition penalty to 1.2 to reduce repetition.

The results, shown in Table 3, indicate that CID+ performs significantly better than DoLa with raw prompting. When CoT is applied, DoLa outperforms CID on GSM8K, MAWPS, and MultiArith. However, DoLa struggled on the SingleEq dataset, where CID consistently improved over the baseline. These findings suggest that while DoLa shows strong performance in certain scenarios, CID demonstrates greater stability across datasets.

**Experiments on Social IQa**  We add another task, Social IQa (Sap et al., 2019), that is not arithmetic reasoning task. We apply CID, CID+ and DoLa to Mistral-Nemo-Instruct on the Social IQa dataset and compare with the original decoding. For DoLa, we adopt the recommended settings for short-answer tasks suggested by the authors: applying DoLa to high layers and setting the repetition penalty to 1.2 to reduce repetition in the generated text. The results are shown in Table 3. We can see that CID and CID+ consistently improve over the original decoding by large gaps. CID is better than DoLa with raw prompts and worse with CoT prompts.

Table 3: Results of apply CID/CID+ and DoLa (Chuang et al., 2023) to Mistral-Nemo-Instruct on arithmetic reasoning datasets and Social IQa dataset (Sap et al., 2019).

| Prompt | Method | GSM8K | MAWPS | MultiArith | SingleEq | Social IQa |
|--------|--------|-------|-------|------------|----------|------------|
| Raw | Orig. | 13.19 | 67.23 | 28.67 | 79.33 | 24.77 |
| | DoLa | 16.00 | 65.13 | 25.50 | 47.91 | 44.93 |
| | CID | 19.71 | 68.49 | 28.00 | 79.72 | **45.80** |
| | CID+ | **45.26** | **71.43** | **48.00** | **84.06** | 28.30 |
| CoT | Orig. | 69.29 | 77.31 | 81.50 | 87.01 | 17.09 |
| | DoLa | **77.63** | **84.03** | **95.17** | 46.41 | **44.37** |
| | CID | 64.82 | 76.05 | 83.00 | 87.40 | 38.54 |
| | CID+ | 62.09 | 83.61 | 83.67 | **87.99** | 24.51 |

