# OpenReview forum: "Unveiling Causal Relationships Among Candidate Output Tokens in Large Language Models: Towards Interpretability and Control"
_ICLR.cc/2025/Conference — Submitted to ICLR 2025_

### Official Review · Reviewer_LYVs · 2024-11-03

**Soundness:** 2
**Presentation:** 2
**Contribution:** 2
**Rating:** 5
**Confidence:** 4

**Summary:**

The paper gives a method  to improve generation in LLMs by exploring causal relationships among candidate output tokens. The authors propose that certain tokens called "cause tokens" activated in early layers of the model causally influence the logits of  "effect tokens" that appear in later layers. To identify these relationships they introduce the Critical Layer Ablation (CLA) heuristic which selectively removes layers to observe their impact on token logits. The authors develop the Causally-Informed Decoding (CID) algorithm which adjusts token probabilities by decreasing the probability of cause tokens and increasing that of effect tokens (aiming to produce more accurate outputs across multiple models). Results show that CID (and CID+) improves reasoning capabilities demonstrating the potential of causally guided decoding for improved language generation.

**Strengths:**

1) The authors define and empirically validate a method to identify “cause tokens” and “effect tokens” during the generation process which is computationally cheaper than uses Peter-Clark (PC) algorithm and this is an interesting approach for control over LLM outputs.

2) The mathematical formulations and causal analysis methods are built on established principles such as CPDAGs and causal discovery through perturbations and using the PC algorithm for initial causal discovery analysis making the approach sound.

**Weaknesses:**

1) The paper’s experimental validation centers on arithmetic reasoning tasks. Arithmetic datasets may not fully capture the complexity of causal dependencies present in broader natural language tasks. Is there any specific reason only arithmetic tasks are considered?

2) In the CID algorithm, why did you choose to adjust logits by simply adding or subtracting a constant value (h) for cause and effect tokens? Were other interventions, such as scaling logits (by some factor) considered? Additionally, adjusting by fixed increments may not account for varying levels of causal influence between tokens. There are no details on how this value (h) is selected. It is only mentioned that CID+ uses a more aggressive set of hyper-parameter configuration.

3) The paper describes using the PC algorithm to detect causal relationships but does not explain the algorithm's workings for their setting. Lack of detail makes it challenging for readers to understand how it is applied. I would suggest the authors to provide a lot more detail on this either in the main paper or in the appendix.

4) Looking at Algorithm 1 describing CLA (specifically lines 7-10): The current logic adds a pair (i,j) to the set of causal relationships only if token j is no longer among the top candidates after ablating the critical layer for token i. This implies that only a drop in j's logit (removal from the top candidates) would count as evidence of a causal relationship from i to j. But this does not allow completeness as if token k's logit increases significantly after ablating the critical layer for token i, this could also indicate a strong causal dependency but it wouldn’t be captured by the current condition. Rather than relying solely on j dropping out of T'  you could calculate the absolute change in j's logit after ablation and use a threshold to determine significance. This weakness is important as the current algorithm is not considering (potentially) a large number of causal pairs due to this.

5) There is an absence of baselines to compare results with across all datasets. Authors should considering comparing their results with alternate causal mediation analysis methods (like ROME- Rank-One Model Editing) or other improved decoding methods which have results on arithmetic datasets like (DoLa - They contrast the differences in logits to improve generation).  Currently there are no other baselines in the paper making it hard to judge how well CID performs.

**Questions:**

1) It is possible that the final set generated by the Critical Layer Ablation (CLA) algorithm could contain both (i,j) and (j,i) (where i,j are tokes) as cause-effect pairs.The tokens i and j can have a mutual influence on each other's logits, such that ablating the critical layer for token i affects token j,and ablating the critical layer for token j affects token i. This could lead to identifying both (i,j) and (j,i) as causal pairs (bidirectional relationship). It might not always precisely capture the true causal direction, especially in complex models like LLMs where token dependencies can be complex. How is this being addressed? Are there any preventative measures to either a) not consider such pairs or b) perform some additioal post processing to determine the true/optimal causal direction?

2) While defining the normalization layer L(v), you have utilized a scaling constant ($\gamma$) but its purpose and how its value are set are unclear. It would help to clarify whether it is used to stabilize logits, adjust their scale, or serve another function, and whether it is constant or varies by model or layer. Additionally, can you provide some insight into how its value is chosen?

---

> ### Author Response · Authors · 2024-11-28
> **Response to Reviewer LYVs (1/3)**
>
> We sincerely appreciate the reviewer for the time and effort invested in evaluating our work and for providing insightful and constructive comments. We have made every effort to address your concerns in the responses below. If there are any remaining questions or issues, please feel free to let us know.
>
> ----------------
> [**Weakness 1**] The reason we used arithmetic reasoning datasets is that many prior works on reasoning and decoding have conducted experiments on these datasets. However, we acknowledge that we overlooked other downstream tasks. To address your concern, we apply CID and CID+ to Mistral-Nemo-Instruct on the Social IQa dataset and compare with the original decoding and DoLa decoding. The results are shown in the table below. We can see that **CID and CID+ consistently improve over the original decoding by large gaps**. CID is better than DoLa with raw prompts and worse with CoT prompts. We have included these results in the Appendix B of the revised manuscript.
>
> | Social IQa | Orig. | DoLa  | CID   | CID+  |
> |:----------:|:-----:|-------|-------|-------|
> |     Raw    | 24.77 | 44.93 | 45.80 | 28.30 |
> |     CoT    | 17.09 | 44.37 | 38.54 | 24.51 |
>
>
> --------------
> [**Weakness 2.1 – Why simply adding or subtracting**] Thanks for pointing this out. Since the logit value appears in the exponent of the weight when calculating the actual sampling probability, **adding a value to the logit effectively corresponds to scaling up the weight for that logit by a factor**. While other interventions, such as scaling logits by a factor, could be considered, our empirical results show that this straightforward approach effectively improves performance. The value of $h$ is a hyperparameter selected based on empirical tuning. We would like to respectfully emphasize that CID is intended as a heuristic technique rather than a method designed for optimality. Its purpose is to **empirically support our hypothesis that causal relationships between candidate tokens can be leveraged to improve the decoding process**.
>
> ----------------
> [**Weakness 2.2 – Configuration of CID and CID+**] We apologize for not explaining in detail how CID and CID+ are different and their specific configurations. CID can be controlled by changing the values of two hyperparameters:
>
> * $d$: the number of tokens with largest logits that will be considered in CLA. Selecting a larger $d$ will result in more cause-effect token pairs to be selected by CLA, and thus more tokens are subject logit changes in CID.
>
> * $h$: the logit change applied to cause and effect tokens detected by CLA. A larger $h$ will alter the token distribution for word prediction more aggressively.
>
> CID+ has a more aggressive configuration than the CID algorithm. Specifically, CID has $(d, h) = (2,5)$ and CID+ has $(d, h) = (5, 10)$. We have included the explanation and the specific configurations in the revised manuscript.

---

> ### Author Response · Authors · 2024-11-28
> **Response to Reviewer LYVs (2/3)**
>
> ---------------
> [**Weakness 3**] We have briefly discussed in lines 231-234 how the PC algorithm is applied to extract cause-effect token pairs in our setting. We apologize that the discussion may not be clear enough, Here we provide more details:
>
> Given an LLM and an input,
>
> 1. We repeatedly perturb the LLM to generate $k=1000$ samples of logit values for the candidate tokens, $\\{s_1,s_2,\ldots,s_k\\}$. Each time,the LLM is perturbed by applying Bernoulli random scalars with success probability 0.95 for the layers as described in Section 3.1, effectively removing some of the transformer layers.
>
> 2. We then apply the PC algorithm to the generated samples $\\{s_1,s_2,\ldots,s_k\\}$ with Fisher’s z independent test and a significance level of 0.9999. This is done using the causal-learn package[1] as footnoted on page 5. The PC algorithm outputs a causal graph between the tokens.
>
> 3. Finally we convert the source and destination tokens of each directed edge of the causal graph to a cause-effect pair.
>
> We hope this provides enough details for understanding the application of the PC algorithm in our setting. We have added the description below in Appendix A in the paper.
>
> *[1] Causal-learn: Causal discovery in python. Zheng, Yujia and Huang, Biwei and Chen, Wei and Ramsey, Joseph and Gong, Mingming and Cai, Ruichu and Shimizu, Shohei and Spirtes, Peter and Zhang, Kun.*
>
> --------------
> [**Weakness 4 – CLA logic**] We appreciate your suggestion to enhance the CLA algorithm by considering significant increases in token logits after layer ablation. While our current implementation focuses on tokens that drop out of the top candidates, incorporating other indicators of causal influence could improve the identification of causal pairs. As noted in our general response, we recognize that CLA is a heuristic and consider improving it an intriguing topic for future research.
>
> ----------------
> [**Weakness 5**] Thank you for your valuable suggestion. While other causal mediation analysis methods do not provide decoding algorithms directly applicable to our context, we agree that comparisons with alternative improved decoding methods, such as DoLa[1], are highly informative. Following your suggestion, we applied DoLa to the Mistral-Nemo-Instruct model across all four datasets used in our paper. We adopted the recommended settings for long-answer reasoning tasks, such as GSM8K, as suggested by the authors of DoLa: applying DoLa to lower layers and setting the repetition penalty to 1.2 to reduce repetition in DoLa decoding.
>
> The results, shown in the table below, indicate that CID+ performs significantly better than DoLa with raw prompting. When CoT is applied, DoLa outperforms CID on GSM8K, MAWPS, and MultiArith. However, DoLa struggled on the SingleEq dataset, where CID consistently improved over the baseline. These findings suggest that **while DoLa shows strong performance in certain scenarios, CID demonstrates greater stability across datasets**. We appreciate your suggestion, as it has helped provide a more comprehensive comparison. We have included these results in the Appendix B in the revised manuscript.
>
> |   Method   | GSM8K | MAWPS | MultiArith | SingleEq |
> |:----------:|:-----:|-------|------------|----------|
> | Raw Prompt |       |       |            |          |
> |    Orig.   | 13.19 | 67.23 | 28.67      | 79.33    |
> |    DoLa    | 16.00 | 65.13 | 25.50      | 47.91    |
> |     CID    | 19.71 | 68.49 | 28.00      | 79.72    |
> |    CID+    | 45.26 | 71.43 | 48.00      | 84.06    |
> | CoT Prompt |       |       |            |          |
> | Orig.      | 69.29 | 77.31 | 81.50      | 87.01    |
> | DoLa       | 77.63 | 84.03 | 95.17      | 46.41    |
> | CID        | 64.82 | 76.05 | 83.00      | 87.40    |
> | CID+       | 62.09 | 83.61 | 83.67      | 87.99    |
>
> *[1] DoLa: Decoding by Contrasting Layers Improves Factuality in Large Language Models. Yung-Sung Chuang, Yujia Xie, Hongyin Luo, Yoon Kim, James Glass, Pengcheng He. ICLR 2024.*

---

> ### Author Response · Authors · 2024-11-28
> **Response to Reviewer LYVs (3/3)**
>
> -----------
> [**Question 1**] This is an insightful question. We have also observed this phenomenon and believe it depends on the design of the heuristic. **It is important to note that CLA is not intended to be a rigorous causal discovery algorithm but rather a fast heuristic for identifying causal pairs efficiently**. An interesting outcome of our study is that this heuristic method does indeed find causal pairs, and the results are statistically significant across many different LLMs, as demonstrated in Section 4.2. Currently, our approach is to ignore such bidirectional pairs in the results. While this may not always capture the true causal direction, we believe the heuristic still serves its purpose effectively in many cases. In the future, we plan to explore alternative approaches or additional post-processing steps to refine the identification of causal directions further.
>
> ----------------
> [**Question 2**] $L(v)$ represents a standard root mean square layer normalization, a commonly used technique for rescaling standardized inputs in various LLM architectures. The parameter $\gamma$ is a learnable scaling factor that is updated during training and remains fixed during inference. As a result, there is no manual selection or adjustment of $\gamma$ in our setting. Additionally, $\gamma$ is not shared across layers, meaning its value can vary from one layer to another.

---

> > ### Comment · Reviewer_LYVs · 2024-12-02
> >
> > Thank you for your responses. I acknowledge that I have read your responses. Given that a new baseline and some of my concerns have been addressed I have updated my score. However, the paper is still scored at 5 (marginally below the acceptance threshold) due to weaknesses 2 and 5 mentioned my review along with question 1.

---

### Official Review · Reviewer_bhjj · 2024-11-04

**Soundness:** 3
**Presentation:** 3
**Contribution:** 2
**Rating:** 6
**Confidence:** 4

**Summary:**

The paper presents:
1. a methodology to find out causal dependencies amongst different output tokens of the vocabulary.
2. Critical Layer Abblation (CLA): A methodology to find critical layers for any token (layer that impacts the logits of a token the most) and using it to deduct potential causal dependencies.
3. Causally Informed Decoding (CID): A decoding algorithm that modifies the autoregressive decoding and improves it for reasoning tasks.

**Strengths:**

- Causal Dependency Analysis: A novel method to find out causal dependencies amongst different output tokens of the vocabulary. It is simple and effective and experiments show that it is able to get reasonable result on causal and effect token deduction.
- Critical Layer Abblation (CLA): A methodology to find critical layers for any token (layer that impacts the logits of a token the most) and using it to deduct potential causal dependencies. Experiments presented on GSM8K.
- Causally Informed Decoding (CID): A decoding algorithm that modifies the autoregressive decoding and improves it for reasoning tasks. High boost in metrics for some models (Gemma 2b andMistral-Nemo)

**Weaknesses:**

- Results are not that significant for CLA in Figure 3. Except for Gemma-2-2B, most of the data points are quite close to y=x and if the blue circles denote the significance interval, most of them don't seem statistically significant. Can the authors weigh in more on why they believe this is good compared to some baseline?
- Results for CID are very mixed as well. While some models do see a lot of jump in their metrics, some do not. Also, it is not clear how CID+ differs from CID and what does more aggressive set of hyperparameters mean. Do the authors have some suggestions on when no-CID, CID, or CID+ should be used based on the dataset or is it just empirical?

**Questions:**

See weakness section.

---

> ### Author Response · Authors · 2024-11-28
> **Response to Reviewer bhjj (1/2)**
>
> We sincerely thank reviewer bhjj for taking the time to review our work and for offering valuable and constructive feedback. We hope that our responses below address all your concerns clearly and thoroughly. Should you have any additional questions or need further clarification, please do not hesitate to let us know.
>
> --------------
> [**Weakness 1 – CLA significance**] We thank the reviewer for pointing this out. As noted in our general response, while some data points in Figure 3 are close to the y=x line, the confidence regions lying above this line indicate statistical significance for multiple models.
>
> --------------
> [**Weakness 2.1 – How CID+ differs from CID**] We apologize for not explaining in detail how CID and CID+ are different and their specific configurations. CID algorithm can be controlled  by changing the values of two hyperparameters:
>
> * $d$: the number of tokens with largest logits that will be considered in CLA. Selecting a larger $d$ will result in more cause-effect token pairs to be selected by CLA, and thus more tokens are subject logit changes in CID.
>
> * $h$: the logit change applied to cause and effect tokens detected by CLA. A larger $h$ will alter the token distribution for word prediction more aggressively.
>
> CID+ has a more aggressive configuration than the CID algorithm. Specifically, CID has $(d, h) = (2,5)$ and CID+ has $(d, h) = (5, 10)$. We have included the specific configurations in the revised manuscript.
>
> ---------------
> [**Weakness 2.2 – Mixed CID results**] We thank the reviewer for pointing out that the CID results seemed mixed. We can observe in Table 2 that most cases where CID or CID+ fails to improve the original decoding are with Gemma-2-9b-it. Therefore, we specifically investigated Gemma-2-9b-it and made some interesting observations that can provide some insights to understand the results.
> We noticed that the texts generated by Gemma-2-9b-it **were formatted well without specifically being prompted**. We take one example from GSM8K to show this. Our prompt is
>
>
> *Given a question, please provide the final answer in the following format: "The answer is [a number here]."\nQuestion: Edgar eats 18 pretzels a day. If his brother eats 1/2 as many, how many does his brother eat in a week?\n Answer:*
>
> The answer generated by Gemma-2-9b-it is:
>
>
> *Here's how to solve the problem:*
>
> * *Find the brother's daily pretzel intake: 18 pretzels / 2 = 9 pretzels*
>
> * *Calculate the brother's weekly pretzel intake: 9 pretzels/day * 7 days/week = 63 pretzels*
>
> * *The answer is 63.*
>
> We observed that Gemma-2-9b-it answered most math questions by starting with “Here’s how to solve the problem:” and following with bulleted steps, **even though we did not prompt it to generate in this format or apply CoT**. This can also be evidenced by the fact that CoT did not help Gemma-2-9b-it at all as shown in Table 2.
>
> We conjecture that this phenomenon can be attributed to how the LLM is instruction-tuned. If the model has been aligned to generate text in specific format, **changing the token distribution as we do in CID will make the generation deviate from the format and thus generate texts of lower quality**.
>
> We would like to emphasize that this is merely a conjecture based on the observations from existing experiments. We are actively investigating this phenomenon and look forward to reporting our findings in future work.

---

> ### Author Response · Authors · 2024-11-28
> **Response to Reviewer bhjj (2/2)**
>
> [**Weakness 2.3 – When to apply which**] Based on our observations above and results reported in Table 2, our suggestion on when to apply which method is:
>
> * When the LLM is small, CID+ is preferred as the reasoning ability of the model is limited and explicit causal reasoning through CID will be helpful.
>
> * When the LLM is larger or it is used with CoT, CID is preferred.
>
> * If the output of the LLM is well formatted without being specifically prompted to do so, it is better to leave the decoding unchanged.

---

### Official Review · Reviewer_zKHP · 2024-11-04

**Soundness:** 3
**Presentation:** 2
**Contribution:** 2
**Rating:** 3
**Confidence:** 4

**Summary:**

The paper introduces Causally-Informed Decoding (CID), a method to improve text generation in language models by prioritizing “effect tokens” over “cause tokens”. Using the Critical Layer Ablation (CLA) heuristic, the authors identify causal relationships among tokens, which CID then leverages to adjust token probabilities during decoding.

**Strengths:**

1. The authors introduce an approach to understanding token dependencies within language models by framing the generation process as a causal structure among tokens, which could have implications for interpretability and controlled text generation.
2. The authors propose a method to efficiently identify causal relationships among tokens, required for real-time demands during decoding.
3. The empirical results show that their approach may improve reasoning capabilities in certain contexts, even if results vary by model and dataset.

**Weaknesses:**

The paper’s premise, proposing that lowering the probability of cause tokens while boosting the probability of effect tokens will improve text generation quality, is questionable. This skepticism mainly stems from the following points:
1. **Bias Analysis** (Section 3.2): In this section, the authors evaluate the robustness of their causal analysis methodology. To do so, they sample from a Bernoulli distribution with varying probabilities $p$ (i.e. the probability of skipping a layer). They claim that higher $p$ values yield Markov equivalence classes that are increasingly similar. However, this observation is intuitive, as fewer skipped layers yield more similar models which in turn lead to more similar outputs. Thus, the conclusion that $p = 0.95$ is closer to $p = 0.9$  than to $p = 0.85$ is self-evident and only offers limited insight into the causality claims presented.
2. **Empirical Validation of CLA** (Section 4.2): In this section, the authors compare the "ground truth" cause-effect pairs derived from the Markov equivalence class with those identified by their CLA. However, in the ROC scatter plot, the fact that CLA predictions are close to $x = y$ suggests that the identified cause-effect pairs do not align well the Markov equivalence class. This observation would imply that the CLA method is not functioning as intended, although the authors claim that "CLA’s predictions are statistically significant across LLMs".
3. **The CID Algorithm** (Section 4.3): In general, the claim that adjusting the probabilities of cause and effect tokens improves text generation quality lacks support. Although Figure 1 shows an example where an effect token gives a correct answer, there’s no guarantee this will always happen. In some cases, the cause token could yield the correct answer, and the effect token could lead to an error. Without more theoretical evidence, the idea that prioritizing effect tokens enhances quality remains unconvincing.
4. **Experiments** (Section 4.4): The paper would benefit from a more detailed explanation of the experimental setup (e.g. specifying what the authors mean by "a more aggressive set of hyper-parameter configuration")

**Questions:**

1. Is there theoretical evidence supporting the claim that prioritizing effect tokens consistently improves text quality across tasks?
2. How do the authors interpret the bias and robustness analyses in Sections 3.2 and 3.3?
3. How feasible is CID in terms of speed and computational demands?

**Details Of Ethics Concerns:**

I don't have any ethical concerns.

---

> ### Author Response · Authors · 2024-11-28
> **Response to Reviewer zKHP (1/2)**
>
> We sincerely thank Reviewer zKHP for the efforts in reviewing our work and for providing constructive comments. We hope this response clarifies all your concerns, as outlined below. Please let us know if you have any further questions or additional concerns.
>
> -------------------------
> [**Weakness 1**] We appreciate your observation regarding the bias analysis. Our main argument in Section 3.2 is that as we introduce varying degrees of bias (by controlling the Bernoulli distribution of layer deletion probability), the causal relationships among candidate tokens remain similar. This is evidenced by the statistical similarity of the Markov equivalence classes we constructed.
>
> While the reviewer's observation is accurate and intuitive—that the model outputs (logit values) tend to resemble those of the full model as the probability of deleting a layer approaches zero—we want to clarify that causal relationships are not determined by the output values themselves but rather by how the outputs are influenced by **changes in the model**. This influence is not directly reflected in the similarity of logits but is instead reflected in the changes caused by the deletion of layers. **Whether these changes tend to be similar as the deletion probability approaches zero is not clear**. Consequently, the increasing similarity of logits does not contradict our analysis in Section 3.2. We acknowledge that our original explanation may have been unclear and potentially misleading, and we have revised Section 3.2 in the updated manuscript to address this point more effectively.
>
> ------------------------
> [**Weakness 2**] We appreciate the reviewer's insightful comments on the Empirical Validation of CLA in Section 4.2. We acknowledge that CLA is a heuristic and not a formal causal discovery method. However, as highlighted in our general response, as long as the confidence regions in the ROC scatter plot lie above the y=x line, the results are statistically significant. This indicates that CLA's identified cause-effect pairs align with the ground truth from the PC algorithm more than would be expected by chance.
>
> Moreover, **we stress that CLA is a heuristic and not a formal technique for causal discovery**. It is intended to **quickly and approximately** identify causal pairs. The interesting outcome of our study is that this heuristic method does indeed find causal pairs, and the results are statistically significant across many different LLMs. This suggests that despite its heuristic nature, CLA is effective in practical applications.
>
> --------------------
> [**Weakness 3 and Question 1**] We acknowledge that no inference algorithm, including our CID algorithm, can guarantee to always output the correct token. Similar to widely used practical inference methods like top-k and top-p sampling, our CID algorithm is designed to **enhance performance without guarantees of correctness in every instance**. Despite this, these algorithms are highly valued for their ability to improve the quality of generated text in practice. Our empirical results on reasoning benchmarks demonstrate that the CID algorithm effectively improves model performance, validating its practical utility.
>
> Regarding the suggestion for more theoretical evidence, we appreciate the feedback and understand the concern. While our causal analysis provides valuable insights and serves as a rigorous foundation for the CID algorithm, we acknowledge that **the effectiveness of CID is primarily evidenced by our benchmark results**. The empirical results from our experiments demonstrate the practical effectiveness of prioritizing effect tokens during decoding. We believe our paper makes two main contributions: (i) **conducting a rigorous causal discovery analysis** that offers theoretical insights into the relationships among tokens, and (ii) demonstrating the effectiveness of the CID algorithm through **empirical results on reasoning benchmarks**. It is also common in machine learning research to place significant value on empirical results, as they provide concrete evidence of an approach's effectiveness. These contributions together support the utility and validity of our approach.

---

> ### Author Response · Authors · 2024-11-28
> **Response to Reviewer zKHP (2/2)**
>
> --------------------------
> [**Weakness 4**] Thank you for pointing out the ambiguity in the description of the algorithm setup. CID algorithm can be controlled  by changing the values of two hyperparameters:
>
> $d$: the number of tokens with largest logits that will be considered in CLA. Selecting a larger $d$ will result in more cause-effect token pairs to be selected by CLA, and thus more tokens are subject logit changes in CID.
>
> $h$: the logit change applied to cause and effect tokens detected by CLA. A larger $h$ will alter the token distribution for word prediction more aggressively.
>
> CID+ has a more aggressive configuration than the CID algorithm. Specifically, CID has $(d, h) = (2,5)$ and CID+ has $(d, h) = (5, 10)$. We have included the explanation and the specific configurations in the revised manuscript in Section 4.4.
>
> -----------------------
> [**Question 2**] We investigate the impact of introducing bias by adding perturbations to the LLM, recognizing that such perturbations could potentially alter the causal relationships among tokens. It is important to study **whether these causal relationships remain consistent when the perturbations are small**, as significant changes could undermine the validity of our causal analysis.
> Our findings indicate that as the perturbations become smaller, the causal relationships we identify remain similar. This suggests that the causal structures are robust to small perturbations added to the LLM. By demonstrating the robustness of these relationships under minor biases, we provide evidence that the **causal connections are inherent to the model and not merely artifacts of the perturbations introduced**.
>
> ------------------
> [**Question 3**] The CID algorithm, in practice, shares the same complexity with the CLA heuristic. They first require one single pass of inference to obtain the initial candidate token logits. Then for the top-k candidate tokens, it requires to run inference with a dropped layer on each token pair to find their approximate causal-effect relation. Denote the single pass complexity as O(pass), the time complexity of the CID and CLA heuristic is O($k^2$ pass),
>
> Consider in practice $k$ is set as a very small number (e.g., 3) and **one could control the frequency to activate the CID algorithm**; the overall time complexity of CID remains within the same order of magnitude as standard inference.
>
> We do observe that CID may spend more time answering a question compared to the standard inference method. This is not due to the complexity of CID itself, but rather because CID increases the lengths of the response. The intuition is that, by using 'effect tokens' such as 'while' instead of directly answering 'yes', the response generated by CID contains more elaboration.

---

> > ### Comment · Reviewer_zKHP · 2024-12-01
> >
> > Thank you for the response. I am still concerned about the effectiveness of your *"heuristic"* algorithm. As you point out, the CLA is not a *"formal technique"*, so you *"place significant value on empirical results"*. However, the empirical results indicate that the performance of CLA in identifying causal relationships is nearly random. Additionally, the original decoding methods (naïve baseline) outperform your approach in 15 out of 48 experiments, which undermines the strength of your empirical claims. For these reasons, I maintain my initial rating. I believe the paper would benefit from stronger theoretical justification and evidence.

---

> > > ### Author Response · Authors · 2024-12-01
> > >
> > > Thank you for your continued engagement with our work and for sharing your concerns. We value your feedback and would like to address your points in detail.
> > >
> > > ---
> > >
> > > **Regarding the empirical performance of CID:**
> > >
> > > We kindly disagree with your argument that "the original decoding methods (naïve baseline) outperform your approach in 15 out of 48 experiments, which undermines the strength of your empirical claims." We would like to clarify that for models where CLA has shown statistical significance (namely Gemma-2-2b-it, Llama-3.2-3B-Instruct, Llama-3.1-8B-Instruct, Yi-1.5-9B-Chat, and Mistral-Nemo-Instruct), the CID algorithm performs worse than the baseline in only 7 out of 40 cases. Moreover, **it is worse than the baseline by more than 1% in only 3 out of these 40 cases**. We believe **this demonstrates empirical significance** and suggests that our approach consistently outperforms or matches the baseline in the majority of cases.
> > >
> > > Additionally, the **statistical significance of CLA can be tested without true labels of decoding data**. Thus, in practice, one can simply avoid using CID on models such as Gemma-2-9b-it, for which CLA is not effective. Therefore, we kindly disagree that the performance of CID undermines the strength of our empirical claims.
> > >
> > >
> > > ---
> > >
> > > **On the effectiveness of CLA in identifying causal relationships:**
> > >
> > > You expressed concern that "the performance of CLA in identifying causal relationships is nearly random." We acknowledge that CLA was not effective on Gemma-2-9b-it. However, **CLA showed statistical significance for multiple other models,** including Gemma-2-2b-it, Llama-3.2-3B-Instruct, Llama-3.1-8B-Instruct, Yi-1.5-9B-Chat, and Mistral-Nemo-Instruct, **indicating that it is not performing at random** but is capturing meaningful causal relationships. Furthermore, as we have mentioned above, for the models where CLA has shown statistical significance, **CID is worse than the baseline by more than 1% in only 3 out of 40 cases**.
> > >
> > > ---
> > >
> > > **Concerning the need for stronger theoretical justification and evidence:**
> > >
> > > We appreciate your suggestion for a stronger theoretical foundation. We believe that the **causal discovery presented in Section 3 serves as a solid basis for our experimental motivation**. If there are specific areas where you feel additional theoretical development is necessary, we would be grateful for more actionable feedback so we can address them appropriately.
> > >
> > > ---

---

> > > > ### Comment · Reviewer_zKHP · 2024-12-02
> > > >
> > > > Thank you for the response. Regarding your empirical claims:
> > > > 1. Your main results (Table 2) include six different language models, four datasets, and two settings (Raw and CoT), resulting in a total of 48 individual experiments. Among these, the baseline *Orig.* achieves the best performance in 15 experiments, accounting for about one-third of the total.
> > > > 2. Your results on the effectiveness of CLA in identifying causal relationships (Figure 3) show that each model has a True Positive Rate (TPR) below 50%, with two out of the five models having a TPR below 20%. While the models may perform statistically significantly better than random, this does not imply that their overall performance is good.
> > > >
> > > > Given the lack of theoretical justification for the method, the empirical results presented are ultimately not convincing.

---

> > > > > ### Author Response · Authors · 2024-12-03
> > > > > **Follow-up response to reviewer's comments**
> > > > >
> > > > > Thank you for your continued engagement during the discussion phase.
> > > > >
> > > > > We acknowledge that the CID algorithm does not outperform the original baseline in certain cases and that some models can have low TPR in CLA. However, we respectfully wish to emphasize several points to suggest that these should not be considered **critical issues** warranting a score of "3".
> > > > >
> > > > > 1. As mentioned in our previous response, if we exclude Gemma-2-9b-it—which did not show statistical significance in the CLA test—the CID algorithm performs worse than the baseline in only 7 out of 40 cases, and in only 3 cases does it perform worse by more than 1%. We believe this level of variability is **normal and acceptable**. For example, in Table 1 of the DoLa paper, DoLa performs worse (by over 1%) than the baselines in 3 cases.
> > > > >
> > > > > 2. We have conducted experiments on **multiple** mainstream families of open-source LLMs, which we consider a significant advantage over previous related works such as DoLa, where experiments were conducted only on Llama. This extensive evaluation strengthens the evidence for the efficacy of CID, as it improves performance on most model families. Furthermore, given the diversity among LLM families and the differences in their pre-training and instruction-tuning processes, it would be practically beneficial to tune CLA and CID hyperparameters specifically for each model family. However, to ensure fair comparisons, we have used the same hyperparameters for CLA and CID across all models.
> > > > >
> > > > > 3. Regarding exceptions like Gemma-2-9b-it, we observed interesting text generation behaviors that might explain why CLA and CID do not perform as well on this model. Specifically, Gemma-2-9b-it generates texts with fixed structures even without explicit prompting. In such cases, CoT is also ineffective (see Table 2). You can refer to our response to Reviewer LYVs for a more detailed discussion. We find this observation interesting and plan to investigate it further in future work.

---

### Author Response · Authors · 2024-11-28
**General Response**

## General response 1: Clarifying the contributions of the paper

Our paper makes two primary contributions:

**Causal Discovery on candidate output tokens:** In Section 3, we present a rigorous causal analysis that reveals the existence of cause-effect relationships among candidate tokens in language models. By applying the Peter-Clark (PC) algorithm, we uncover the underlying causal structures that govern token dependencies during the generation process.

**Causally-Informed Decoding Algorithm (CID):** We introduce the CID algorithm, an empirical decoding method that leverages the identified causal relationships to adjust token probabilities during decoding. The Critical Layer Ablation (CLA) and CID heuristics are not meant to be optimal; they should be evaluated based on the accuracy of the decoding algorithm itself to demonstrate an actual application of the causal discovery. While not designed for optimality, these heuristics effectively demonstrate how causal discovery can inform and improve decoding strategies in practice.

## General response 2: Alignment Between CLA and PC Algorithm Results

We acknowledge that CLA is a heuristic designed for efficiency rather than exactness. However, our empirical results indicate that CLA is capable of identifying cause-effect pairs with statistical significance for several models, such as Llama-3.2-3B and Mistral-Nemo. As shown in Figure 3, the confidence regions for these models lie entirely above the y=x line in the TPR vs. FPR plot, indicating strong alignment between CLA's predictions and the ground truth causal relationships identified by the PC algorithm.

For models like Gemma-2-9B, the causal pairs extracted by CLA are not significant, as evidenced by the scatter points lying close to the y=x line. This suggests that CLA is less effective for these models, which is reflected in the diminished performance of CID (see Table 2). Importantly, the effectiveness of CLA may serve as an indicator of CID's performance, providing an early assessment even when ground truth labels for decoding are unavailable.

For other models, including Yi-1.5-9B and Gemma-2-2B, the alignment between CLA and the PC algorithm is statistically significant for either cause tokens or effect tokens, albeit less strong. We recognize that this is due to CLA being a heuristic. We appreciate the reviewers' suggestions on improving CLA and find it an intriguing topic for future research to develop more advanced heuristics for detecting causal pairs.

---

### Meta-Review · Area_Chair_uJvV · 2024-12-20

**Metareview:**

This paper proposes to extract cause-effect relationships among candidate tokens during LLM generation by treating the output tokens as effect tokens which are causally influenced by tokens activated in the earlier layers. The authors perform causal analysis to confirm the cause-effect relationship and propose a novel causally-informed decoding algorithm to manipulate token probabilities during generation, decreasing the effect brought by cause tokens. Experiments demonstrate the advantage of the proposed method in enhancing model reasoning capabilities.

Strengths:
- The proposed idea of exploiting causal relationship among candidate tokens is novel and interesting, which could potentially benefit further studies in controllability.
- The method is novel and supported by sound analysis and theoretical guarantees.
- Empirical results demonstrate the effectiveness of CID for enhancing the reasoning performances across mathematical reasoning datasets.

Weaknesses:
- Most reviewers expressed concerns regarding empirical results which show that the proposed method does not consistently (or at least mostly) outperform existing baselines. From Table 2, CID and CID+ still underperforms Orig. in several experiments, limiting the contributions in real applications.
- The significance of CLA remains uncertain. The ROC plots for CLA do not show a clear statistical significance compared with the $y=x$ baseline.
- The experiments on math reasoning datasets could limit the method's generalizability in other application domains.

**Additional Comments On Reviewer Discussion:**

- Most reviewers raised concerns about the experimental results where the proposed method CID/CID+ is inferior than the original decoding strategy in several experiments. While the authors tried to explain the implication, it is still not fully convincing that the method could significantly benefit model reasoning.
- Reviewers also raised concerns about the statistical significance of the CLA method, given that the ROC plots do not reveal a clear separation from the baseline $y=x$ for some language models. The rebuttal does not seem to convince the reviewers.
- Despite the above two points, the authors have provided further clarifications on experimental settings (such as CID+), additional experiments on another reasoning dataset besides mathematical reasoning and comparison with advanced decoding methods. These additional efforts are helpful in strengthening the contribution of this paper. Nevertheless, the first two limitations are still the main concern when evaluating the significance of this work.

---

### Decision · Program_Chairs · 2025-01-22

Reject